# Socioeconomic inequalities of 3-year survival in formal employees with colorectal cancer between 2012 and 2019 in Colombia

**Daniela Sánchez-Santiesteban**[1,2]*, **Andrés Felipe Patiño-Benavidez**[1,2], **Giancarlo Buitrago**[1,2]

1 Instituto de Investigaciones Clínicas, Facultad de Medicina, Universidad Nacional de Colombia, Bogotá D.C., Colombia, 2 Fundación Cardioinfantil–Instituto de Cardiología, Bogotá D.C., Colombia

* dansanchezsa@unal.edu.co

## Abstract

### Background

Colorectal cancer is a high-burden disease that requires comprehensive multidisciplinary management. In Colombia, despite a healthcare system covering 97% of the population, socioeconomic disparities persist. Lower income levels are associated with decreased survival, potentially due to delays in diagnosis or treatment and a higher probability of advanced staging at diagnosis, These inequities persist even among relatively advantaged populations, such as formal employee who are assumed to have fewer barriers to accessing healthcare services compared to informal workers.

### Objective

This study aimed to assess the association monthly minimum wages (MMW) as a measure of socioeconomic status in three-year survival among formal employees diagnosed with colorectal cancer in Colombia from 2012 to 2019.

### Methods

A retrospective cohort study was conducted using administrative databases that included healthcare and mortality records. Formal employees newly diagnosed with colorectal cancer were identified through diagnostic and oncological procedure codes and were followed for three years from the date of diagnosis or until death. The exposure variable was the legal monthly minimum wage (MMW) at the time of diagnosis, used as a proxy for socioeconomic status, while the outcome variable was three-year survival. Patients were stratified into quartiles based on their MMW. The three-year mortality proportion was calculated for each quartile. To assess survival differences, Cox proportional hazards regression models were applied to estimate adjusted hazard ratios (HRs). Socioeconomic gradients in survival were quantified using the Relative Index of Inequality (RII) and the Slope Index of Inequality (SII).

**Data availability statement:** The following information sources: Single Registry of Enrollees (Registro Único de Afiliación, or RUAF), Unique Affiliation Database (Base de Datos Única de Afiliación, or BDUA), Wages Database (PILA), and Calculation Study of the Capitation Unit Database (Base del Estudio de Suficiencia de la Unidad Por Capitación, or UPC) are administered by the Colombian Ministry of Health and Social Protection. These databases are freely available upon request to the Technology of the Information and Communication Office of the Colombian Ministry of Health and Social Protection through the e-mail: correo@minsalud.gov.co.

**Funding:** Authors DSS and GB were partially supported by a research grant from the NIHR GHPSR researcher-led grant NIHR150067, using UK international development funding from the UK Government to support global health research. The views expressed in this publication are those of the authors and do not necessarily reflect the views of the NIHR or the UK Government.

**Competing interests:** The authors have declared that no competing interests exist.

## Results

The cohort included 1,913 formal employees (mean age: 49.9 years), with 660 deaths (34.5%) recorded over the follow-up period. Patients in the lowest MMW quartile experienced the highest three-year mortality (39.5%) compared to those in the highest quartile (30.7%). After adjusting for confounders, individuals in the highest quartile had a 25% lower risk of death than those in the lowest quartile (aHR: 0.74; 95% CI: 0.59–0.92). The RII indicated a 50% higher risk of death in the lowest income group (RII: 1.50; 95% CI: 1.13–1.99), while the SII revealed an absolute difference of 0.16 deaths per 1,000 individuals (p=0.01).

## Conclusion

Significant income-based disparities in colorectal cancer survival were observed among formal employees in Colombia despite the theoretically equitable healthcare system. These findings underscore the persistent influence of socioeconomic factors on health outcomes, even within populations assumed to have better access to care.

## Introduction

Colorectal cancer is one of the leading causes of cancer-related morbidity and mortality worldwide. In 2020, it ranked as the second most deadly cancer and the third most commonly diagnosed, with an estimated 1.9 million new cases and 935,000 deaths annually [1]. Projections indicate that by 2040, the global burden could rise to 3.2 million new cases and 1.6 million deaths yearly [2]. In Colombia, colorectal cancer was the third leading cause of cancer incidence and mortality in 2021, primarily affecting individuals aged 50 and older, many of whom belong to the formal workforce [3].

In low- and middle-income countries (LMICs) like Colombia, formal employees are often regarded as a relatively advantaged group due to their regular income and access to mandatory social security benefits, including healthcare, pensions, and occupational risk insuranceand access to employer-provided health benefits [4]. This category includes both private and public sector workers, as long as their employment meets these legal requirements. However, this perceived advantage does not entirely shield them from significant barriers to adequate healthcare access, particularly for complex conditions such as colorectal cancer. This disease requires timely and multidisciplinary care, which can be hindered by structural and socioeconomic factors, even among populations with formal employment and health coverage [5,6].

The Colombian health system, comprising contributory and subsidised schemes, covers over 97% of the population and aims to ensure equitable healthcare access regardless of affiliation [7]. Additionally, clinical practice guidelines have been developed to standardise diagnosing and managing diseases like colorectal cancer. Despite these measures, evidence from Colombia reveals persistent socioeconomic disparities in early detection, timely treatment, and survival outcomes, even within systems designed to promote equity [8–10]. Geographic barriers significantly influence healthcare access, with regions like Bogotá and the Central region benefiting from better infrastructure and health outcomes, while the Amazon and Pacific regions face challenges such as difficult terrain, limited transportation, and inequitable healthcare resource distribution. These barriers disproportionately affect remote and disadvantaged populations, leading to delays in accessing healthcare services and resulting in poorer clinical outcomes.

Previous research has highlighted the role of income as a significant determinant of cancer outcomes, including colorectal cancer survival [11]. Studies have shown that individuals with lower income levels often experience delayed access to diagnosis and treatment, leading to more advanced disease stages at presentation and poorer survival rates. These findings persist even in settings with universal healthcare systems, suggesting that broader structural and socioeconomic inequalities significantly impact outcomes. By focusing on a population of formal employees in Colombia, this study provides an opportunity to examine how income disparities within a relatively advantaged group may still influence survival outcomes, thereby contributing to the broader understanding of health inequities in cancer care.

This study aims to assess the association between using monthly minimum wages (MMW) as a measure of socioeconomic status in three-year survival among formal employees diagnosed with colorectal cancer in Colombia from 2012 to 2019. By focusing on a population assumed to have relatively uniform access to healthcare, this analysis highlights the persistent influence of income disparities on health outcomes. It contributes to the broader understanding of social determinants of health.

## Methods

### Data source and study population

This retrospective cohort study analysed administrative databases to examine the entire population of formal employees in Colombia who were newly diagnosed and treated for colorectal cancer between January 1, 2012, and December 31, 2019 [12,13]. The cohort was constructed using the Base for the Study of Capitation Unit Sufficiency (UPC) to identify patients meeting criteria established by an electronic algorithm based on diagnostic and oncological procedure codes. Since ICD-10 codes in the UPC database may indicate confirmed or suspected diagnoses, electronic algorithms are essential to identify individuals with specific conditions accurately. Colorectal cancer cases were defined as patients with at least three diagnostic codes recorded in different months and at least one oncological procedure code (e.g., chemotherapy, radiotherapy, or surgery). This algorithm has been validated and used in previous studies [14–17]. The list of ICD-10 and CUPS codes used in the algorithm is available in the S1 File.

The final study sample included all adults who met the colorectal cancer diagnosis algorithm criteria during the study period and were actively employed at the time of diagnosis. Cohort entry was defined as the date of the last recorded ICD-10 code required to meet the eligibility criteria, with follow-up extending for three years or until death, whichever occurred first. Data was accessed on August 10, 2024. This study was approved by the Faculty of Medicine Ethics Committee at the Universidad Nacional de Colombia (Approval Number: 019–179).

This study utilised four primary administrative databases, anonymised and linked using unique identifiers by the Ministry of Health (MoH). The UPC includes comprehensive records of healthcare service utilisation for individuals within the health system, capturing information on the type of services provided, associated ICD-10 codes, costs, service dates, municipalities, and healthcare providers. This database supports risk-adjusted capitated payment calculations and is populated by the largest insurers within the contributory scheme, covering approximately 19 million individuals (88% of this scheme's population nationwide). It is highly standardised and reports minimal underreporting of health service utilisation [12]. The Unique Enrollees Database (BDUA) is the government registry for tracking demographic and enrollment information of all health system beneficiaries across contributory, subsidised, and other schemes. It provides data on enrolment status, age, sex, insurer, and affiliation scheme [18]. The Death Registry (RUAF) compiles mortality data from death certificates,

including location, date, cause of death, and sociodemographic characteristics (e.g., sex, age, marital status, education level, occupation, and ethnicity). International assessments have confirmed the reliability of RUAF data, with 91% of deaths registered through death certificates as of 2016 [13,19]. Finally, the Wages Database (PILA), managed by the MoH, contains detailed records on payroll contributions to health, pensions, and social security and precise information on formal employee wages. This database enables analysis of employment trends and wage distribution. While self-employed individuals self-report income, formal employee wage data is aggregated and reported by employers, ensuring higher data quality for this group [20]. These datasets, widely used in national studies [14,16,21–24], were provided to the Clinical Research Institute of the Universidad Nacional de Colombia for this research.

## Study variables

This study examined socioeconomic inequalities in 3-year survival among colorectal cancer patients. The primary exposure variable was the patient's socioeconomic position, measured as the formal wage earned during cohort entry, as recorded in the PILA database. This variable was expressed in units of legal monthly minimum wages (MMW) and further stratified into quartiles to facilitate analysis. The primary outcome variable was 3-year survival time, defined as the period from the cohort entry date to the date of death or the end of the 3-year follow-up period, whichever occurred first. The date of death was obtained from death certificates in the RUAF database.

We included sociodemographic and clinical variables measured during colorectal cancer diagnosis to control for potential confounders. Sociodemographic variables included age, sex, insurer, and geographic region (Colombia is divided into 32 departments). Clinical variables included the Charlson Comorbidity Index (CCI), validated for use in Colombia by Oliveros et al. [24], derived from ICD-10 codes. The CCI considered the presence of the following conditions: acute myocardial infarction, congestive heart failure, peripheral vascular disease, stroke, dementia, chronic pulmonary disease, connective tissue disease, peptic ulcer disease, liver disease, diabetes, complications of diabetes, cancer, metastatic cancer, paraplegia, renal disease, severe liver disease, and HIV. Based on the oncological services used during diagnosis and treatment, we classified the cancer stage as local, locally advanced, or advanced.

Sociodemographic and clinical variables were measured prior to cohort entry, specifically at the time of the first recorded ICD-10 code that initiated the colorectal cancer identification algorithm. This approach ensures that all covariates were captured before the patients entered the cohort, avoiding the inclusion of variables measured during follow-up that could act as potential mediators. By defining cohort entry as the date when the eligibility criteria were fully met, we also minimize biases in the measurement of covariates that could arise if data were collected during the period between the initial diagnostic codes and the fulfillment of the algorithm.

## Statistical analysis

We performed descriptive statistics on the sociodemographic characteristics of formal employees with colorectal cancer. Continuous variables with normal distributions were summarised using means and standard deviations, while continuous variables with non-normal distributions were summarised using medians and interquartile ranges. Categorical variables were summarised using absolute and relative frequencies. The descriptive analysis was stratified according to quartiles of wages, the classification of monthly income into quartiles was based on the dataset distribution to ensure an equitable representation of the population and avoid subjective decisions in defining boundaries. This approach is widely

used in socioeconomic studies and facilitates comparisons across regions and time periods. We described the three-year mortality proportion by quartiles of wages with 95% confidence intervals. Using semiparametric Cox regression models, we estimated unadjusted and adjusted hazard ratios (HRs) and 95% CIs for three-year survival. Variables used for model adjustment included sex, age, year, insurer, geographic region, CCI, cancer stage, and year of diagnosis. Statistical and graphical tests were performed to evaluate the assumption of proportionality.

We estimated the Relative Index of Inequality (RII) and the Slope Index of Inequality (SII) to quantify the socioeconomic gradient in absolute and relative terms, following the structured regression framework proposed by Moreno-Betancur et al. These indices allow for a continuous and standardized assessment of socioeconomic disparities, minimizing bias and enhancing comparability across populations. As these indices can only be estimated for socioeconomic characteristics with at least three categories, they were identified for the quartile of MMW at the month of first diagnosis of colorectal cancer. Following the recommendations of Moreno-Betancur et al., we used Poisson regression models to estimate the RII as the incidence rate ratio between the individual with the best socioeconomic position and the one with the worst. After conducting the Poisson regression, the SII was calculated from the absolute difference in the predicted mortality rates per 1,000 adults for the two groups. All indices were adjusted with the variables previously mentioned in the survival models. All analyses were conducted using Stata v.18 MP (Universidad Nacional de Colombia license). This article followed the STROBE (Strengthening the Reporting of Observational Studies in Epidemiology) guidelines to ensure transparency and thoroughness in reporting the findings. Additionally, an artificial intelligence language model, specifically OpenAI's GPT-4, assisted with style corrections and enhanced the text's clarity and coherence during the writing process.

## Results

Between January 1, 2012, and December 31, 2019, 1,950 formal employees were diagnosed with colorectal cancer. After excluding 41 records (2.10%) due to incomplete data on variables such as age and sex, the final cohort consisted of 1,909 individuals who were followed for three years or until death, whichever occurred first. The flowchart in Fig 1 provides additional details on the selection process. Among the final cohort, 1,162 individuals (60.87%) were male, and the mean age at diagnosis was 49.88 years (SD: 10.58). The majority of patients (1,229; 64.38%) were diagnosed with locally advanced colorectal cancer, and 76.43% had a CCI of 2 at the time of diagnosis. Most cases were concentrated in Bogotá (740 cases; 38.76%) and the Central region (554 cases; 29.02%).

Table 1 presents detailed sociodemographic and clinical characteristics of the cohort. The distribution of monthly minimum wages (MMW) was stratified into quartiles, with the 25th, 50th, and 75th percentiles corresponding to 1.00, 1.31, and 2.34, respectively. MMW values ranged from 1.00 to 24.81. Median MMWs varied significantly across regions, with the Orinoquía y Amazonías region reporting the highest median MMW (5.63) and the Atlántica region the lowest (1.24).

A total of 718 deaths were observed during the follow-up period, corresponding to a three-year mortality proportion of 37.61%. The regions with the highest mortality rates were Atlántica (44.60%) and Pacífica (42.49%), exceeding the overall cohort mortality proportion. Mortality proportions stratified by MMW quartiles revealed significant differences. Patients in the lowest MMW quartile had the highest three-year mortality (42.41%; 95% CI: 38.10–46.85), while those in the highest quartile had the lowest (33.68%; 95% CI: 29.57–38.05). Fig 2 illustrates the three-year mortality rates across MMW quartiles, showing a transparent socioeconomic gradient.

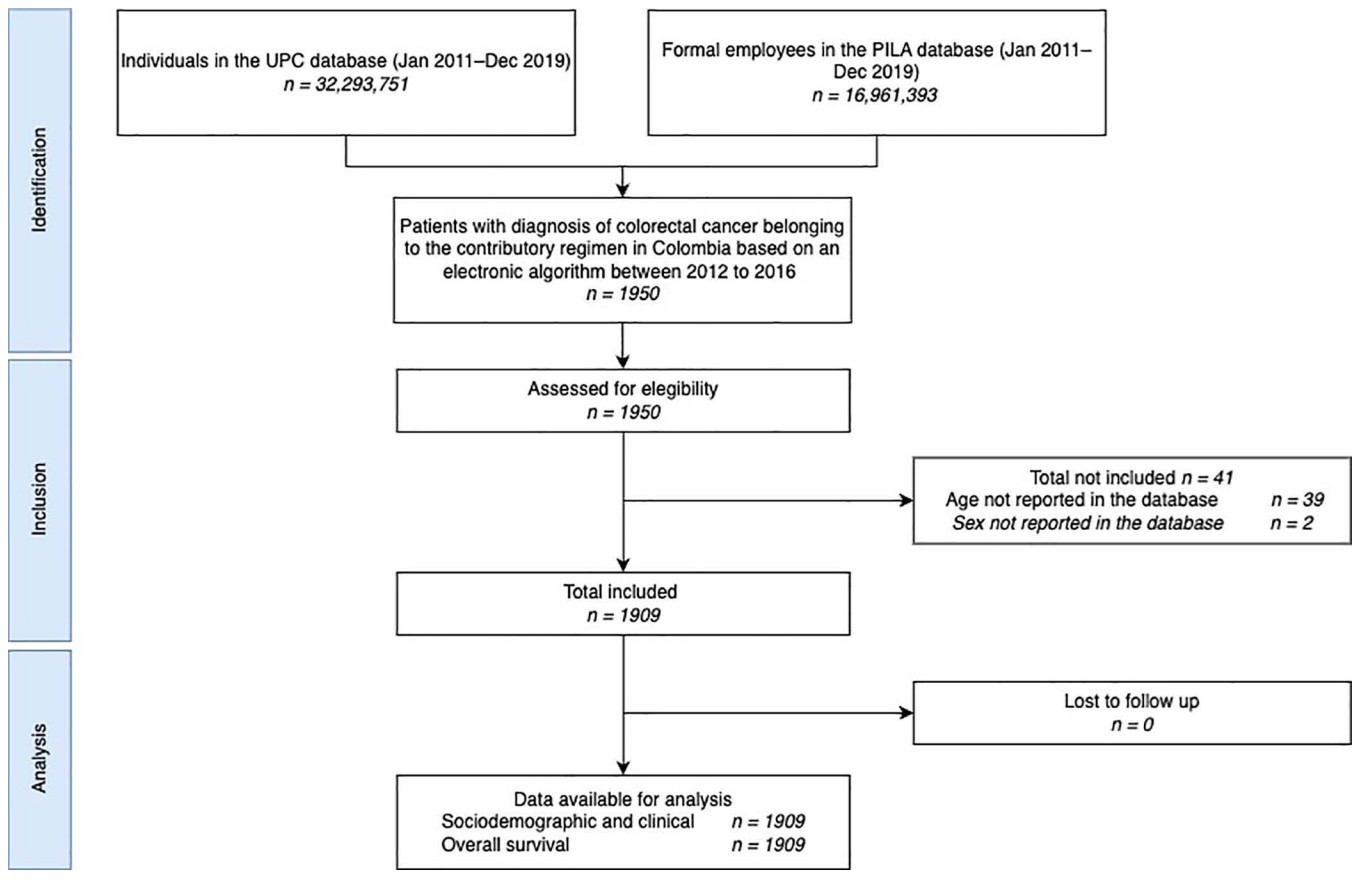

**Fig 1. STROBE flow chart.** UPC: Base for the Study of Capitation Unit Sufficiency Database; PILA: Wages Database.

The risk of death within three years of a colorectal cancer diagnosis significantly increased with lower MMW, as shown in both crude and adjusted hazard ratios. p25 of Survival time for the cohort was 1.72 years. After adjusting for age, sex, cancer stage, CCI, region, insurer, and year of diagnosis, formal employees in the highest MMW quartile had a 25% lower risk of death compared to those in the lowest quartile (adjusted HR:0.75; 95% CI: 0.59–0.91). Table 2 summarises the unadjusted and adjusted HRs for each MMW quartile. At the same time, Fig 3 displays survival curves from the multivariate models, highlighting a dose-response relationship: lower MMWs were associated with reduced survival.

The calculated RII and SII further supported these findings, showing significant differences by MMW even after adjustment for key covariates (Table 3).

## Discussion

This population-based study focused on formal employees diagnosed with colorectal cancer in a low- and middle-income country (LMIC), specifically Colombia. Over seven years, patients were followed for three years to assess survival outcomes. To our knowledge, this is the first study in an LMIC targeting a theoretically advantaged population due to their formal employment status and the associated health benefits. Our findings reveal significant disparities in three-year survival rates based on income, even within this ostensibly protected group. Despite being part of a health system designed to ensure equitable access to healthcare services, our results underscore that income remains a critical determinant of survival,

**Table 1. Baseline characteristics.**

| | | MMW quartiles | | | |
|---|---|---|---|---|---|
| | Total sample | Q1 | Q2 | Q3 | Q4 |
| | N=1,909 | N=488 | N=463 | N=483 | N=475 |
| Age, mean (SD) | 49.88 (10.58) | 51.73 (10.96) | 48.61 (11.21) | 48.25 (10.08) | 50.87 (9.62) |
| Categorized age, n (%) | | | | | |
| Under 40 years old | 358 (18.75%) | 73 (14.96%) | 104 (22.46%) | 107 (22.15%) | 74 (15.58%) |
| 40 to 60 years old | 1,257 (65.85%) | 314 (64.34%) | 293 (63.28%) | 327 (67.70%) | 323 (68.00%) |
| Over 60 years old | 294 (15.40%) | 101 (20.70%) | 66 (14.25%) | 49 (10.14%) | 78 (16.42%) |
| Male, n (%) | 1,162 (60.87%) | 297 (60.86%) | 280 (60.48%) | 283 (58.59%) | 302 (63.58%) |
| Stagea, n (%) | | | | | |
| Local | 152 (7.96%) | 34 (6.97%) | 42 (9.07%) | 39 (8.07%) | 37 (7.79%) |
| Locally advanced | 1,229 (64.38%) | 316 (64.75%) | 291 (62.85%) | 316 (65.42%) | 306 (64.42%) |
| Metastatic | 528 (27.66%) | 138 (28.28%) | 130 (28.08%) | 128 (26.50%) | 132 (27.79%) |
| CCI median (p25 - p75) | 2.00 (2.00-2.00) | 2.00 (2.00-2.00) | 2.00 (2.00-2.00) | 2.00 (2.00-2.00) | 2.00 (2.00-3.00) |
| Categorised CCI, n (%) | | | | | |
| 2 | 1,459 (76.43%) | 367 (75.20%) | 360 (77.75%) | 383 (79.30%) | 349 (73.47%) |
| 2-3 | 298 (15.61%) | 79 (16.19%) | 73 (15.77%) | 67 (13.87%) | 79 (16.63%) |
| ≥5 | 152 (7.96%) | 42 (8.61%) | 30 (6.48%) | 33 (6.83%) | 47 (9.89%) |
| Regionb, n (%) | | | | | |
| Atlántica | 157 (8.22%) | 40 (8.20%) | 39 (8.42%) | 40 (8.28%) | 38 (8.00%) |
| Bogotá DC | 740 (38.76%) | 158 (32.38%) | 189 (40.82%) | 199 (41.20%) | 194 (40.84%) |
| Central | 554 (29.02%) | 164 (33.61%) | 134 (28.94%) | 126 (26.09%) | 130 (27.37%) |
| Oriental | 183 (9.59%) | 40 (8.20%) | 45 (9.72%) | 41 (8.49%) | 57 (12.00%) |
| Pacífica | 273 (14.30%) | 86 (17.62%) | 56 (12.10%) | 76 (15.73%) | 55 (11.58%) |
| Other departments | 2 (0.10%) | 0 (0.00%) | 0 (0.00%) | 1 (0.21%) | 1 (0.21%) |

Data are means (SD). [a]Stage categories were defined by initial treatment: early (only surgery or surgery-radiotherapy), locally advanced (surgerychemotherapy or chemotherapy-surgery-radiotherapy), and advanced (radiotherapy-chemotherapy, only radiotherapy, or only chemotherapy). [b]Region: Atlantica: Atlántico, Bolivar, Cesar, Cordoba, Magdalena, La Guajira, Sucre, and San Andres; Central: Antioquia, Caldas, Huila; Risaralda, Tolima, and Quindio; Oriental: Boyaca, Caquetá, Santander, Norte de Santander, Arauca, Meta, Casanare; and Pacífica: Valle del Cauca, Cauca, Choco, and Nariño; Orinoquía y Amazonia: Guainía, Amazonas, Guaviare, Vaupés and Vichada.

Abbreviations: MMW: Monthly Minimum Wages; SD: standard deviation; p25: percentil 25; p75: percentil 75; CCI: Categorized Charlson Index.

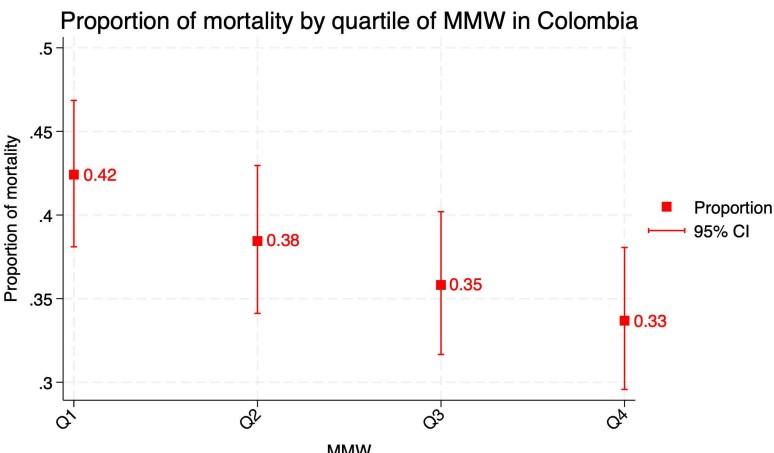

**Fig 2. Three-year mortality proportion of formal employees with colorectal cancer per quartile of MMW.** MMW: Monthly Minimum Wages.

**Table 2. Unadjusted and adjusted HR of overall survival of formal employees with colorectal cancer per quartile of MMW.**

| MMW quartiles | Unadjusted HR of overall survival | | | Adjusted HR of overall survival[a] | | |
|---|---|---|---|---|---|---|
| | HR | 95% CI | p | HR | 95% CI | p |
| Q1 | ref | ref | ref | Ref | Ref | Ref |
| Q2 | 0,88 | 0.72 - 1.07 | 0,21 | 0,87 | 0.71 - 1.10 | 0,19 |
| Q3 | 0,82 | 0.67 - 0.99 | 0,04 | 0,80 | 0.65 - 0.99 | 0,04 |
| Q4 | 0,75 | 0.61 - 0.92 | 0,01 | 0,73 | 0.59 - 0.91 | 0,01 |

[a]The HR was adjusted for Sex, Stage of Cancer at the moment of diagnosis, Region, CCI, insurer, and year of diagnosis.

Abbreviations: MMW: Monthly Minimum Wages; HR: Hazard Ratio; 95% CI, 95% confidence interval; p: p-value, Q: Quartiles.

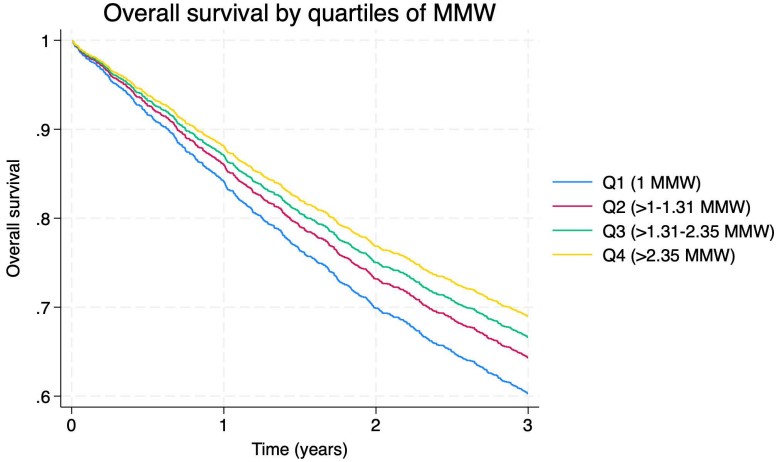

**Fig 3. Overall three-year survival of formal employees with colorectal cancer per quartile of MMW.** MMW: Monthly Minimum Wages.

**Table 3. Relative and Slope Index of inequality (RII & SII) in overall mortality of patients with colorectal cancer belonging to the contributory regime in Colombia by MMW.**

| | Index | 95% CI | p-value |
|---|---|---|---|
| RII | | | |
| Overall mortality | 1.46 | 1.13–1.90 | 0.01 |
| SII | | | |
| Overall mortality (per 1.000 adults) | 0.17 | 0.05–0.28 | 0.01 |

Relative and Slope Index of inequality were calculated by controlling by sex, age, stage of diagnosis, Charlson Comorbidity Index, geographic region, insurer and year.

Abbreviations: RRI: Relative Index of inequality; SII: Slope Index of inequality.

challenging the assumption of universal access and highlighting persistent inequalities in health outcomes.

These findings align with research conducted in both high-income countries and LMICs, where substantial disparities in cancer survival related to socioeconomic status have been documented, even in systems striving for equitable healthcare access [25]. Numerous studies have demonstrated the impact of income inequalities on colorectal cancer survival, consistently reporting lower survival rates among individuals with lower incomes [26–30]. Disparities have also been identified across other dimensions, including geographic location, residential

socioeconomic status, education level, race, and health insurance coverage [9,31–34]. Our findings are further supported by studies evaluating income-based inequalities using indicators such as the SII and the RII, which consistently show significant differences in outcomes based on these factors [35,36]. This study contributes to this growing body of evidence by highlighting that income plays a crucial role in survival outcomes, even within a relatively advantaged group, such as formal employees, who theoretically face fewer healthcare access barriers.

It is important to note that income disparities often intersect with other forms of inequality, such as differences in education, working conditions, geographic barriers, and the quality of healthcare services. For example, lower-income individuals may reside in areas with limited access to specialised healthcare or face longer wait times for advanced treatments, even in systems that aim to provide equitable healthcare access. These interconnected factors likely contribute to the observed disparities in survival outcomes, even in settings where primary healthcare access is ostensibly uniform. Beyond access to specialised care, other structural barriers—such as the quality and continuity of care, navigation through the healthcare system, and the availability of multidisciplinary teams—may also significantly influence survival outcomes.

This study has several strengths. The reliability of the data is a crucial advantage, as it includes comprehensive information on income and formal employee status, with wage data mandatorily reported by employers to the PILA database under national regulations. Furthermore, the RUAF death registry provides extensive coverage, documenting over 90% of deaths nationwide, making it a robust resource for investigating healthcare inequalities. These databases, including the UPC and BDUA, have been widely used in previous studies, further supporting their validity as valuable tools for research. However, some limitations should be considered. Patient identification relied on algorithms using procedural and ICD-10 codes, which, while necessary for administrative databases, could lead to misclassification. Additionally, although administrative databases provide reliable data on healthcare utilisation, they lack detailed clinical variables that could offer a more nuanced understanding of cancer characteristics. To mitigate this limitation, we staged the disease based on procedure codes related to cancer treatment, supported by clinical expertise.

Finally, this study focuses on a relatively privileged population in terms of health status and income. The income-related disparities identified here may be even more pronounced in other population groups, such as informal employees or the unemployed. Moreover, survival differences could be exacerbated in countries with greater overall inequality. Thus, while our findings are significant, they may underestimate the true extent of socioeconomic disparities in colorectal cancer outcomes across broader or more disadvantaged populations, given that lower-income workers who are less likely to use healthcare services may be underrepresented in our service-dependent database. Although we accounted for several confounding factors, the potential for residual confounding due to unmeasured or unknown variables cannot be eliminated. Further studies are needed to explore the impact of social inequalities on health outcomes in populations with poorer socioeconomic conditions and more significant barriers to adequate healthcare access.

## Supporting information

**S1 File.  List of ICD-10 and CUPS codes is used in the electronic algorithm for colorectal cancer.**
(DOCX)

**S1 Table.  Coefficients of survival regression.**
(DOCX)

## Acknowledgments

We thank Colombia's Ministry of Health and Social Protection for providing the administrative databases that made this study possible. We also thank the Clinical Research Student Group at the Faculty of Medicine of the National University of Colombia for their valuable contributions. This study's preliminary findings were presented and discussed within this group, enriching the interpretation and scope of our results. The School of Medicine at Universidad Nacional de Colombia supported the study.

## Author contributions

**Data curation:** Andrés Felipe Patiño-Benavidez, Giancarlo Buitrago.

**Formal analysis:** Daniela Sánchez-Santiesteban, Andrés Felipe Patiño-Benavidez, Giancarlo Buitrago.

**Investigation:** Daniela Sánchez-Santiesteban, Andrés Felipe Patiño-Benavidez.

**Methodology:** Daniela Sánchez-Santiesteban, Andrés Felipe Patiño-Benavidez, Giancarlo Buitrago.

**Software:** Daniela Sánchez-Santiesteban, Andrés Felipe Patiño-Benavidez, Giancarlo Buitrago.

**Writing – original draft:** Daniela Sánchez-Santiesteban, Andrés Felipe Patiño-Benavidez, Giancarlo Buitrago.

**Writing – review & editing:** Daniela Sánchez-Santiesteban, Andrés Felipe Patiño-Benavidez, Giancarlo Buitrago.

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
