## [Decision Letter · Decision Letter 0]

16 Jan 2025

PONE-D-24-55586Socioeconomic inequalities of 3-year survival in formal employees with colorectal cancer between 2012 and 2019 in ColombiaPLOS ONE

Dear Dr. Sánchez-Santiesteban,

Thank you for submitting your manuscript to PLOS ONE. After careful consideration, we feel that it has merit but does not fully meet PLOS ONE’s publication criteria as it currently stands. Therefore, we invite you to submit a revised version of the manuscript that addresses the points raised during the review process.

**Please ensure that all comments for a better article were made.**

We look forward to receiving your revised manuscript.

Kind regards,

Alejandro Botero Carvajal, MD

Academic Editor

PLOS ONE

**Journal Requirements:**

DSS and GB were partially supported by a research grant from the NIHR GHPSR researcher-led grant NIHR150067, which used UK aid from the UK Government to support global health research. The funders had no role in study design, data collection and analysis, decision to publish, or preparation of the manuscript.

We thank Colombia's Ministry of Health and Social Protection for providing the administrative databases that made this study possible. We also thank the Clinical Research Student Group at the Faculty of Medicine of the National University of Colombia for their valuable contributions. This study's preliminary findings were presented and discussed within this group, enriching the interpretation and scope of our results.

The School of Medicine at Universidad Nacional de Colombia supported the study. Additionally, this research was partially funded by the NIHR GHPSR researcher-led grant NIHR150067, which used UK aid from the UK Government to support global health research.

DSS and GB were partially supported by a research grant from the NIHR GHPSR researcher-led grant NIHR150067, which used UK aid from the UK Government to support global health research. The funders had no role in study design, data collection and analysis, decision to publish, or preparation of the manuscript.

Reviewers' comments:

Reviewer's Responses to Questions

**Comments to the Author**

1. Is the manuscript technically sound, and do the data support the conclusions?

Reviewer #1: Yes

Reviewer #2: Partly

Reviewer #3: Partly

Reviewer #4: Partly

2. Has the statistical analysis been performed appropriately and rigorously? 

Reviewer #1: Yes

Reviewer #2: Yes

Reviewer #3: Yes

Reviewer #4: Yes

3. Have the authors made all data underlying the findings in their manuscript fully available?

Reviewer #1: Yes

Reviewer #2: Yes

Reviewer #3: Yes

Reviewer #4: No

4. Is the manuscript presented in an intelligible fashion and written in standard English?

Reviewer #1: Yes

Reviewer #2: Yes

Reviewer #3: Yes

Reviewer #4: Yes

5. Review Comments to the Author

**Reviewer #1:**  This manuscript evaluated the survival after a colorectal cancer diagnosis among formal employees among different wages strata as a proxy for socio-economic status.

The study is well conducted and statistically analysis well done.

I just have one question for the authors, why did you choose to stop the follow-up after 3 years for everyone instead of just following until the end of the death registry/last date of update of this registry? I would like to see the median and IQR of the follow-up time for deaths, to see if they occurred more close to the start of the follow-up or closer to the 3 years. It could be added to the table 1 I suppose.

I also noticed few typos (example figure 2 « quartil » instead of « quartile »).

**Reviewer #2:**  On line 87, the study's objective is described as assessing socioeconomic inequalities in three-year survival among formal workers with colorectal cancer. However, the study primarily outlines the socioeconomic characteristics of these patients, particularly income, age, sex, and place of residence in Colombia. It does not provide sufficient evidence to support the inference that the socioeconomic factors studied directly determine survival outcomes. The statistical analysis focuses on the association between income and survival, based on two outcomes: three-year survival or death. I suggest revising the objective to better align with the results, which are significant deserve emphasis. For example: "To analyze the association between income and survival in patients with colorectal cancer in a country with low socioeconomic development."

Regarding the geographic location in Colombia, the characteristics of the regions and the geographic barriers faced by each were not described. Without this context, it is unclear to the readers how these factors could influence the study results. Moreover, the functioning of the country's health system was not addressed. If the authors are referring to geographic barriers, more detail is needed, as these barriers can indeed affect access to care and may ultimately influence patient survival. However, the study does not provide specifics about the characteristics of these regions.

Additionally, in reference to the statement "Although administrative databases provide reliable data on healthcare utilization, they lack detailed clinical variables that could offer a more nuanced understanding of cancer characteristics," I believe that important clinical details, such as the type of cancer and its histopathology (which may indicate more aggressive lesions, leading to shorter survival), are not available. This is a crucial factor to consider when analyzing mortality.

**Reviewer #3:**  Using a retrospective cohort study, the authors look at socioeconomic inequalities in three-year survival of employees diagnosed with colorectal cancer in Colombia, 2012-2019. The paper reports that individuals in the highest quartile had a 25% lower risk of death than those in the lowest quartile, and concludes that significant income-related differences in colorectal cancer survival were observed among employees in the Colombian formal sector.

The research questions explored in this study – if repeated in a number of robust studies – can be of significance for the improvement of healthcare delivery and outcomes by providing insights and recommendations for healthcare practitioners and policymakers on necessary support processes, aligned with attainable patient-centered goals.

With that in mind, this reviewer has the following to remark:

1. Abstract

1.1 The text of this section can be improved.

1.2 The authors could consider adding a short background paragraph.

2. Methods

2.1 How did the authors address the possibility of selection bias and detection bias?

2.2 The authors mention that they used an artificial intelligence language model to enhance clarity and coherence.

It must be noted that the text appears to be in need of some editorial attention to improve readability. There seems to have been too much reliance on AI. It is evident from a number of observations, one of which is that the phrase “this study” was repeated many times.

While there are areas where AI can support academic writing, too much reliance may backfire. The use of AI must be coupled with an appropriate level of human oversight and control. It is advisable, therefore, that the authors edit/revise their manuscript to ensure its overall coherence.

3. Results & Discussion

3.1 Even though the authors have accounted for a number of potential confounding factors, there is always the possibility of residual confounding due to unmeasured or unknown factors.

Have the authors been mindful of the latter?

3.2 If so, why haven't they mentioned it in the discussion section?

3.3 The authors write, “Thus, while our findings are significant, they may underestimate the true extent of socioeconomic disparities in cancer outcomes across broader or more disadvantaged populations.”

This sentence may be perceived as a generalized statement – something that the authors need to avoid. Their study looked at colorectal cancer, not at cancer development in general. It is prudent to stick to the type of cancer the authors state to have examined.

I hope this review is helpful and wish the authors the very best with their research!

**Reviewer #4: ** Summary

The authors conducted a retrospective cohort study in Colombia to quantify the impact of social in-equailities on survival in colorectal cancer related deaths.

Major points

- While there's no problem of requiring patients to have multiple codes to confirm their eligibility to the study, indexing patients on a timepoint based on them meeting a condition in the future can be problematic and can open the door for immortal time bias. This is actually clear in the survival curve being flat at the beginning of the follow up. I recommend the authors to re-define the cohort entry date to be at the latest date in which the eligibility criteria was satisfied.

- Lines 143:152

- This goes into the first issue as well. It's not clear from the text whether these coavariates were measured during the follow-up or before the follow-up. Based on the text, the follow-up started from the first occurrence of the cancer identification algorithm code. This means that many of those covariates might've been measured during the follow-up, which can lead to overadjutment due to adjustment for potential mediators.

- I am wondering whether the exposure variable was turned into quartiles based on the dataset for analysis or other outside resource. The issue of relying on the data is that such classification scheme is highly arbitrary and might not lead to good replication in other studies. I would recommend the authors to rely on previous literature to define the bounadries of their exposure variable classes.

- Why didn't the authors adjust for healthcare utilization and concurrent medications?

- Another source of concern is that secondary type of exposure that the authors used to show social inequality. After we saw non-significant difference in HRs across different income categories with a straightforward exposure (MMW), the authors resorted to two indices to show that indeed inequality exists. The authors need to provide a clear justification in the switch of their choice of the exposure. Do they believe these indices are more reflective of social inequality? What is the evidence for this?

Minor points

- Is 'formal employee' the same thing as government employee?

- Line 169 needs a citation.

- In the flowchart, I am wondering if the numbers in the very first boxes represent records or patients. I understand that patient can have multiple records. For this flowchart, the authors should be consistent and should report numbers related to the patients not records.

- In table 2, the comma of the hazard ratios should be corrected to a decimal point.

6. PLOS authors have the option to publish the peer review history of their article (what does this mean? ). If published, this will include your full peer review and any attached files.

**Do you want your identity to be public for this peer review?** For information about this choice, including consent withdrawal, please see our Privacy Policy .

Reviewer #1: No

Reviewer #2: **Yes: ** NAIDHIA ALVES SOARES FERREIRA

Reviewer #3: No

Reviewer #4: No

---

## [Author Response · Author response to Decision Letter 0]

6 Feb 2025

Dear Editors and Reviewers,

We sincerely thank you for the valuable feedback provided on our manuscript. Your thoughtful comments and suggestions have enhanced the work's clarity, rigour, and overall quality. Below, we address each comment individually, providing detailed explanations and outlining the specific changes made to the manuscript.

Reviewer #1:

This manuscript evaluated the survival after a colorectal cancer diagnosis among formal employees among different wages strata as a proxy for socio-economic status. The study is well conducted and statistically analysis well done.

Comment 1: I just have one question for the authors, why did you choose to stop the follow-up after 3 years for everyone instead of just following until the end of the death registry/last date of update of this registry?

Response 1: Thank you for your valuable comment. We chose a uniform 3-year follow-up period because we calculated 3-year mortality proportions, a clinically relevant endpoint widely reported in colorectal cancer studies. In addition, adopting the same observation window for all participants ensures that these proportions remain comparable across the entire cohort and facilitates direct comparisons of our findings with other research using the same 3-year outcome measure.

Comment 2: I would like to see the median and IQR of the follow-up time for deaths, to see if they occurred more close to the start of the follow-up or closer to the 3 years. It could be added to the table 1 I suppose.

Response 2: Thank you for this suggestion. We could not describe the median as deaths only happened in 37.61 % of the population, however, we described p25 of survival time (page 10, line 263-264): “p25 of Survival time for the cohort was 1.72 years”.

Comment 3: I also noticed few typos (example figure 2 « quartil » instead of « quartile »).

Response 3: Thank you for pointing this out. The manuscript has been carefully reviewed for typographical errors, and corrections have been made, including the text in Figure 2.

Reviewer #2:

Comment 1: On line 87, the study's objective is described as assessing socioeconomic inequalities in three-year survival among formal workers with colorectal cancer. However, the study primarily outlines the socioeconomic characteristics of these patients, particularly income, age, sex, and place of residence in Colombia. It does not provide sufficient evidence to support the inference that the socioeconomic factors studied directly determine survival outcomes.

Response 1: Thank you for your insightful comment. We have restructured the introduction's fourth paragraph to support in a better way, the potential association between income and colorectal cancer survival. The updated paragraph reads as follows (page 4, lines 102-109): “Previous research has highlighted the role of income as a significant determinant of cancer outcomes, including colorectal cancer survival. Studies have shown that individuals with lower income levels often experience delayed access to diagnosis and treatment, leading to more advanced disease stages at presentation and poorer survival rates. These findings persist even in settings with universal healthcare systems, suggesting that broader structural and socioeconomic inequalities significantly impact outcomes. By focusing on a population of formal employees in Colombia, this study provides an opportunity to examine how income disparities within a relatively advantaged group may still influence survival outcomes, thereby contributing to the broader understanding of health inequities in cancer care.”

Comment 2: The statistical analysis focuses on the association between income and survival based on two outcomes: three-year survival or death. I suggest revising the objective to better align with the results, which are significant and deserve emphasis. For example: "To analyse the association between income and survival in patients with colorectal cancer in a country with low socioeconomic development."

Response 2: Thank you for your valuable comment. We agree with your suggestion and have revised the objective accordingly to better align with the results and emphasise their significance. The updated objective now reads (page 5, lines 111-113): “This study aims to assess the association of monthly minimum wages (MMW) as a measure of socioeconomic status in three-year survival among formal employees diagnosed with colorectal cancer in Colombia from 2012 to 2019.”

Comment 3: Regarding the geographic location in Colombia, the characteristics of the regions and the geographic barriers faced by each were not described. Without this context, it is unclear to the readers how these factors could influence the study results. Moreover, the functioning of the country's health system was not addressed. If the authors are referring to geographic barriers, more detail is needed, as these barriers can affect access to care and may ultimately influence patient survival. However, the study does not provide specifics about the characteristics of these regions.

Response 3: Thank you for your insightful comment. We acknowledge the importance of providing additional context regarding the geographic characteristics of the regions in Colombia, the inequities they face, and the potential influence of these barriers on access to care and patient survival. In response, we have included a more detailed description of the regional disparities, geographic barriers, and their implications for the healthcare system. The following information has been added to the manuscript (page 4, lines 95-100) to enhance clarity and provide a more comprehensive understanding of the study context: “Geographic barriers significantly influence healthcare access, with regions like Bogotá and the Central region benefiting from better infrastructure and health outcomes, while the Amazon and Pacific regions face challenges such as difficult terrain, limited transportation, and inequitable healthcare resource distribution. These barriers disproportionately affect remote and disadvantaged populations, leading to healthcare service delays and poorer clinical outcomes.”

Comment 4: Additionally, about the statement, "Although administrative databases provide reliable data on healthcare utilisation, they lack detailed clinical variables that could offer a more nuanced understanding of cancer characteristics," I believe that important clinical details, such as the type of cancer and its histopathology (which may indicate more aggressive lesions, leading to shorter survival), are not available. This is a crucial factor to consider when analysing mortality.

Response 4: We sincerely appreciate your insightful comment and fully agree on the relevance of clinical variables, such as histopathology, in predicting survival outcomes. While our limitations section highlights the need for studies with more detailed clinical information, your observation underscores the importance of incorporating specific data, such as histopathology, which could shed light on variations in survival based on cancer characteristics. That said, for this study, which focuses on identifying survival differences related to income levels, we consider that histopathological type might be associated with exposures, lifestyle factors, and other variables influenced by income. Thus, it could act as a mediator in the relationship between income and survival. Your comment reinforces the complexity of these interactions and the need for future research to integrate clinical and socioeconomic dimensions.

Reviewer #3:

Using a retrospective cohort study, the authors look at socioeconomic inequalities in three-year survival of employees diagnosed with colorectal cancer in Colombia, 2012-2019. The paper reports that individuals in the highest quartile had a 25% lower risk of death than those in the lowest quartile, and concludes that significant income-related differences in colorectal cancer survival were observed among employees in the Colombian formal sector.

The research questions explored in this study – if repeated in a number of robust studies – can be of significance for the improvement of healthcare delivery and outcomes by providing insights and recommendations for healthcare practitioners and policymakers on necessary support processes, aligned with attainable patient-centered goals.

Comment 1: With that in mind, this reviewer has the following to remark:

1. Abstract

1.1 The text of this section can be improved.

1.2 The authors could consider adding a short background paragraph.

Response 1: Thank you for your thoughtful feedback and valuable suggestions. We agree with your comments regarding the need to improve the clarity of the abstract and include a brief background paragraph. In response, we have revised the abstract to enhance its structure and readability and added a concise background section to provide context for the study. We appreciate your insights, as they have helped strengthen the presentation of our findings.

Comment 2:

2. Methods

2.1 How did the authors address the possibility of selection bias and detection bias?

Response 2: Thank you for this critical question. We hypothesise that a gradient in survival exists favouring workers with higher economic privilege compared to those with lower incomes, likely due to factors such as access to services, timeliness of care, and access to more experienced healthcare facilities, among others. Given that our database records healthcare service utilisation, selection bias may be favouring workers with better health conditions, even within lower-income strata. This selection could make lower-income workers similar to higher-income ones, potentially underestimating the magnitude of the effect and biasing the results toward the null hypothesis of no differences between groups.

To address this concern, we have revised the limitations section by incorporating the following sentence (page 13, lines 340-343): “Thus, while our findings are significant, they may underestimate the full extent of socioeconomic disparities in cancer outcomes among broader or more disadvantaged populations, given that lower-income workers who are less likely to use healthcare services may be underrepresented in our service-dependent database.”

Regarding detection bias, the RUAF mortality registry is highlighted as a strength of our study due to its high coverage and accuracy. This registry is completed by healthcare personnel, and the socioeconomic status of the individuals does not influence its documentation. Therefore, we consider it a reliable and unbiased source for determining patient survival.

Comment 3:

2.2 The authors mention that they used an artificial intelligence language model to enhance clarity and coherence.

It must be noted that the text appears to be in need of some editorial attention to improve readability. There seems to have been too much reliance on AI. It is evident from a number of observations, one of which is that the phrase “this study” was repeated many times.

While there are areas where AI can support academic writing, too much reliance may backfire. The use of AI must be coupled with an appropriate level of human oversight and control. It is advisable, therefore, that the authors edit/revise their manuscript to ensure its overall coherence.

Response 3: We appreciate this valuable feedback regarding the language and coherence of the manuscript. In response to your comment, we have meticulously reviewed the manuscript to improve readability and coherence. Specifically, we addressed repetitive phrases such as “this study” and rephrased sentences where redundancy was noted. Additionally, we have ensured that the overall flow of the text aligns with professional academic standards.

Comment 4:

3. Results & Discussion

3.1 Even though the authors have accounted for a number of potential confounding factors, there is always the possibility of residual confounding due to unmeasured or unknown factors.

Have the authors been mindful of the latter?

Response 4: Thank you for raising this critical point. Despite our efforts to account for key confounders in the analysis, we acknowledge the possibility of residual confounding due to unmeasured or unknown factors. Variables such as lifestyle factors, detailed clinical characteristics, and genetic predispositions, unavailable in our dataset, may influence the observed associations.

To address this, we have emphasised this limitation in the discussion section and underscored the need for further research that integrates both socioeconomic and detailed clinical data to provide a more comprehensive understanding of these interactions. A sentence was added to the discussion section (page 13, lines 343-345) to acknowledge the potential for residual confounding explicitly: “Although we accounted for several confounding factors, the potential for residual confounding due to unmeasured or unknown variables cannot be eliminated.”

Comment 5: 3.2 If so, why haven't they mentioned it in the discussion section?

Response 5: Thank you for your comment. We have explicitly mentioned the possibility of residual confounding in the discussion section

Comment 6: 3.3 The authors write, “Thus, while our findings are significant, they may underestimate the true extent of socioeconomic disparities in cancer outcomes across broader or more disadvantaged populations.” This sentence may be perceived as a generalised statement – something that the authors need to avoid. Their study looked at colorectal cancer, not at cancer development in general. It is prudent to stick to the type of cancer the authors state to have examined.

I hope this review is helpful and wish the authors the very best with their research!

Response 6: Thank you for your insightful comment. We agree that the statement could be interpreted as overly broad and have revised it to refer to colorectal cancer specifically. This adjustment ensures the discussion is accurate and remains within the scope of our study.

The revised sentence in the discussion section now reads (page 13, lines 340-342): “Thus, while our findings are significant, they may underestimate the true extent of socioeconomic disparities in colorectal cancer outcomes across broader or more disadvantaged populations.”

Reviewer #4:

Summary

The authors conducted a retrospective cohort study in Colombia to quantify the impact of social in-equailities on survival in colorectal cancer related deaths.

Major points

Comment 1: While there's no problem of requiring patients to have multiple codes to confirm their eligibility to the study, indexing patients on a timepoint based on them meeting a condition in the future can be problematic and can open the door for immortal time bias. This is actually clear in the survival curve being flat at the beginning of the follow up. I recommend the authors to re-define the cohort entry date to be at the latest date in which the eligibility criteria was satisfied.

Response 1: Thank you for pointing out this critical methodological issue. In response, we have redefined the cohort entry date to ensure alignment with eligibility criteria and to mitigate the risk of immortal time bias. Cohort entry is now defined as the date of the last recorded ICD-10 code required to meet the eligibility algorithm. This ensures that time zero reflects the point at which all criteria were satisfied.

The manuscript now defines cohort entry as follows (page 5, lines 133-135): “Cohort entry was defined as the date of the last recorded ICD-10 code required to meet the eligibility criteria, with follow-up extending for three years or until death, whichever occurred first.”

Following this adjustment, all survival analyses, tables, and figures were updated to reflect the revised cohort entry definition. The updated estimates now provide a more accurate representation of survival outcomes while maintaining the study’s robustness.

This methodological refinement is informed by the work of Hernán et al. [1], who emphasise the importance of aligning cohort entry with eligibility criteria to address biases arising from immortal time. By implementing this approach, we ensure that the analyses are methodologically sound and free from biases that could

---

## [Decision Letter · Decision Letter 1]

19 Mar 2025

Socioeconomic inequalities of 3-year survival in formal employees with colorectal cancer between 2012 and 2019 in Colombia

PONE-D-24-55586R1

Dear Dr. Sánchez-Santiesteban,

We’re pleased to inform you that your manuscript has been judged scientifically suitable for publication and will be formally accepted for publication once it meets all outstanding technical requirements.

Kind regards,

Alejandro Botero Carvajal, MD

Academic Editor

PLOS ONE

Additional Editor Comments (optional):

Reviewers' comments:

Reviewer's Responses to Questions

**Comments to the Author**

1. If the authors have adequately addressed your comments raised in a previous round of review and you feel that this manuscript is now acceptable for publication, you may indicate that here to bypass the “Comments to the Author” section, enter your conflict of interest statement in the “Confidential to Editor” section, and submit your "Accept" recommendation.

Reviewer #1: All comments have been addressed

2. Is the manuscript technically sound, and do the data support the conclusions?

Reviewer #1: Yes

3. Has the statistical analysis been performed appropriately and rigorously? 

Reviewer #1: Yes

4. Have the authors made all data underlying the findings in their manuscript fully available?

Reviewer #1: Yes

5. Is the manuscript presented in an intelligible fashion and written in standard English?

Reviewer #1: Yes

6. Review Comments to the Author

Reviewer #1: Thanks for taking in account all reviewers's comments, great work was done to address all of them. I do not have further comments.

7. PLOS authors have the option to publish the peer review history of their article (what does this mean? ). If published, this will include your full peer review and any attached files.

**Do you want your identity to be public for this peer review?** For information about this choice, including consent withdrawal, please see our Privacy Policy .

Reviewer #1: No

---

## [Editor Report · Acceptance letter]

PONE-D-24-55586R1

PLOS ONE

Dear Dr. Sánchez-Santiesteban,

I'm pleased to inform you that your manuscript has been deemed suitable for publication in PLOS ONE. Congratulations! Your manuscript is now being handed over to our production team.

Kind regards,

on behalf of

Dr. Alejandro Botero Carvajal

Academic Editor

PLOS ONE